# A Hyaluronic Acid Demilune Scaffold and Polypyrrole-Coated Fibers Carrying Embedded Human Neural Precursor Cells and Curcumin for Surface Capping of Spinal Cord Injuries

**DOI:** 10.3390/biomedicines9121928

**Published:** 2021-12-16

**Authors:** Hoda Elkhenany, Pablo Bonilla, Esther Giraldo, Ana Alastrue Agudo, Michael J. Edel, María Jesus Vicent, Fernando Gisbert Roca, Cristina Martínez Ramos, Laura Rodríguez Doblado, Manuel Monleón Pradas, Victoria Moreno Manzano

**Affiliations:** 1Neuronal and Tissue Regeneration Laboratory, Centro de Investigación Príncipe Felipe, 46012 Valencia, Spain; hoda.atef@alexu.edu.eg (H.E.); pbonilla@cipf.es (P.B.); esgire@btc.upv.es (E.G.); aalastrue@cipf.es (A.A.A.); 2Department of Surgery, Faculty of Veterinary Medicine, Alexandria University, Alexandria 22785, Egypt; 3Department of Biotechnology, Universitat Politècnica de València, 46022 Valencia, Spain; 4Unit of Anatomy and Embryology, School of Medicine, Autonomous University of Barcelona, 08193 Barcelona, Spain; Michael.Edel@uab.cat; 5Centre for Cell Therapy and Regenerative Medicine (CCTRM), Harry Perkins Research Institute, University of Western Australia, Perth 6009, Australia; 6International Research Fellow, Victor Chang Cardiac Research Institute, Sydney 2010, Australia; 7Polymer Therapeutics Laboratory, Centro de Investigación Príncipe Felipe, 46012 Valencia, Spain; mjvicent@cipf.es; 8Center for Biomaterials and Tissue Engineering, Universitat Politècnica de València, 46022 Valencia, Spain; nandogisbert@gmail.com (F.G.R.); cris_mr_1980@hotmail.com (C.M.R.); laura.rodriguez.doblado@gmail.com (L.R.D.); mmonleon@ter.upv.es (M.M.P.)

**Keywords:** spinal cord injury, PuraMatrix hydrogel, hyaluronic acid, biomaterials, induced neural progenitor cells, curcumin

## Abstract

Tissue engineering, including cell transplantation and the application of biomaterials and bioactive molecules, represents a promising approach for regeneration following spinal cord injury (SCI). We designed a combinatorial tissue-engineered approach for the minimally invasive treatment of SCI—a hyaluronic acid (HA)-based scaffold containing polypyrrole-coated fibers (PPY) combined with the RAD16-I self-assembling peptide hydrogel (Corning^®^ PuraMatrix™ peptide hydrogel (PM)), human induced neural progenitor cells (iNPCs), and a nanoconjugated form of curcumin (CURC). In vitro cultures demonstrated that PM preserves iNPC viability and the addition of CURC reduces apoptosis and enhances the outgrowth of Nestin-positive neurites from iNPCs, compared to non-embedded iNPCs. The treatment of spinal cord organotypic cultures also demonstrated that CURC enhances cell migration and prompts a neuron-like morphology of embedded iNPCs implanted over the tissue slices. Following sub-acute SCI by traumatic contusion in rats, the implantation of PM-embedded iNPCs and CURC with PPY fibers supported a significant increase in neuro-preservation (as measured by greater βIII-tubulin staining of neuronal fibers) and decrease in the injured area (as measured by the lack of GFAP staining). This combination therapy also restricted platelet-derived growth factor expression, indicating a reduction in fibrotic pericyte invasion. Overall, these findings support PM-embedded iNPCs with CURC placed within an HA demilune scaffold containing PPY fibers as a minimally invasive combination-based alternative to cell transplantation alone.

## 1. Introduction

Spinal cord injury (SCI) treatment remains a significant challenge due to the complexity and dynamic nature of the intrinsic pathological cascades that occur immediately after the primary lesion and progress to the permanent loss of neuronal activity and the creation of anatomical impediments to treatment application. Pre-clinical studies of complementary neuroprotective and neuroregenerative interventions have provided evidence that such approaches can limit the progression and amplification of the initial damage and rescue neurological deficits (as reviewed by Ahuja and Fehlings [1]). Secondary injury damage beginning soon after the initial trauma results in massive neuronal and glial cell death (including oligodendrocytes), prompting demyelination and expanding the loss of efficient neuronal connectivity to additional neuronal tracts. Persistent inflammatory cell infiltration enhanced by the loss of selective vascular permeability also promotes the continual formation of toxic cystic microcavities. Overall, the early interruption of this second wave of damage may reduce injury severity and, therefore, prompt better functional preservation and prognosis.

The formation of a permissive platform bridging the extrinsic inhibitory microenvironment characteristic of the injured spinal area via cell transplantation could afford collateral neuroplasticity and neuronal regeneration [1]. The transplantation of neural progenitor cells derived from human induced pluripotent stem cells (iNPCs) [2] has provided promising results in terms of repair and neuronal regeneration in rodent [3,4] and primate [5,6] models; however, the limited functional and anatomical improvements obtained in rat models (with poor cell survival being a significant contributor) suggest that iNPC transplantation alone would fail to provide sufficient gain of function. Therefore, the ongoing clinical development of cell transplantation approaches requires additional supportive complementary therapeutic strategies.

The harsh host immune response faced by transplanted cells, which limits survival and engraftment in the host tissue, represents another significant therapeutic obstacle; however, embedding or encapsulation using protective agents may minimize this impact [7]. The biodegradable hydrogel PuraMatrix (PM), the commercial name for the RAD16-I self-assembling peptide hydrogel, represents a candidate for nervous tissue engineering given its ability to support cell proliferation and differentiation [8,9,10] and its biocompatible, biodegradable, and conductive nature [10,11]. PM also provides a suitable three-dimensional (3D) microenvironment for iNPCs to promote differentiation into neurons and astrocytes, nerve regeneration, myelination, and axon regrowth across the SCI lesion [12]. Recently, transplantation studies with PM-embedded human NPCs demonstrated enhanced cell survival and differentiation, reduced lesion volume, and improved neurological functions in a model of SCI [13].

Our recent studies demonstrated the synergism of iNPC transplantation with a nanoconjugated form of the anti-inflammatory molecule curcumin (CURC), which provided long-term drug release and an increase in tissue bioavailability thanks to conjugation to a polyacetal polymer via a pH-responsive linker [14]. The intrathecal delivery of CURC and the intramedullary transplantation of iNPCs and human mesenchymal stem cells polarized microglia towards an anti-inflammatory profile, reduced the extent of the injured area, and increased neuronal fiber preservation, thereby providing a more versatile approach for acute SCI treatment [14].

The intramedullary administration of cell therapies has been intensely debated in terms of clinical application [15]. The spinal cord architecture, surrounded by the vertebrae body and the meningeal layers, confers a critical intrinsic anatomical limitation to local treatment. Additionally, invasive interventions in the soft tissue of the spinal cord can lead to additional tissue damage and neurological dysfunction [16]. Thus, the clinical application of therapeutics requires the development of minimally invasive approaches that reduce tissue manipulation. Surface capping of SCI has been proposed as a minimally invasive cell and drug delivery approach [17]. This technique requires the integration of biocompatible, non-toxic, and biodegradable biomaterials to safely deliver and transfer cells compatible with the spinal cord soft tissue anatomy. Hyaluronic acid (HA), a non-sulfated glycosaminoglycan, meets the biocompatibility requirements for soft tissue [18,19,20,21]. We previously discovered that a highly porous HA demilune (crescent-moon-shaped) scaffold containing polylactic acid (PLA) fibers in the internal lumen seeded with NPCs derived from neonatal rat spinal cord tissue successfully preserved neural tissue with minimal cyst and scar formation [17]. The greater hydrophobicity of PLA fibers than HA allows the adsorption of extracellular matrix proteins, which supports integrin-mediated adhesion of cells and the formation of focal adhesions that promote cell survival. Integrins directly activate survival pathways via the phosphoinositide 3-kinase and mitogen-activated protein kinase pathways [22]. Hence, the characteristics of HA and PLA robustly influence cell morphology and adhesive response [23]. As we recently demonstrated, HA demilune scaffolds filled with PLA fibers coated with polypyrrole (PPY) favor axonal growth and the guidance of dorsal root ganglia explants [21]. Consequently, PPY’s application is relevant to biomedical applications, especially in nerve tissue engineering, thanks to its biocompatibility, high electrical conductivity, long-term environmental stability, low cost, and accessible synthesis by chemical or electrochemical polymerization [24,25,26,27]. These features qualify PPY as a potentially crucial factor in neural regeneration approaches [28,29]. It is of note that PPY is an intractable solid with very low mechanical processability, which limits direct application as a substrate with a specific topography [25]; therefore, many approaches employ PPY as a surface coating for polymers with better mechanical processability, thereby exploiting the electrical conductivity of PPY and the mechanical properties of a support polymer [30,31]. In this study, we chose PLA as a support polymer, as its easy mechanical processability allows the production of biocompatible and biodegradable microfibers [32,33].

The present study characterized a novel combinatory cell therapy approach by embedding iNPCs and CURC in a PM hydrogel seeded into a biocompatible capping-like HA scaffold containing PPY-coated PLA guiding fibers. This biocomposite supported minimal invasiveness during implantation and is supported in the acutely injured spinal cord.

## 2. Materials and Methods

### 2.1. Embedding iNPCs and CURC within PM

The iNPCs were generated as previously described [34]. Briefly, human iPSCs were reprogrammed using reprogramming factors (OCT3/4, SOX2, KLF4, LIN28) and a synthetic mRNA coding for CYCLIN D1, which supports genetic stability during the reprogramming process. For iNPC generation, a PSC Neural Induction Medium kit (Life Technologies, Framingham, MA, USA) was employed. After neural induction, iNPC cultures were maintained in growth medium (GM) STEMdiff™ Neural Progenitor Medium (STEMCELL™, Vancouver, BC, Canada) supplemented with 200 U/mL penicillin and 200 μg/mL streptomycin (Lonza, Basel, Switzerland) in standard cell incubation conditions. For the iNPC sub-culture, cells at 80% confluence were detached using TrypLE^TM^ Select (Thermo Fisher Scientific, Agawam, MA, USA). A total of 1 × 10^3^ iNPCs/µL were infected with a lentiviral vector containing GFP (PLL-eGFP) at a multiplicity of infection of 10 for 1 h and then plated with fresh GM. After 72 h of proliferation, the iNPCs were assessed for positive lentiviral transduction.

The PM hydrogel used for embedding iNPCs was prepared at 0.3% by mixing 30 µL pure PM, 50 µL 20% sucrose, and 20 µL dH_2_O and then sonicated for 30 min. A total of 5 µL of 0.3% PM hydrogel was mixed with 5 × 10^4^ iNPCs in 5 µL of sucrose (final PM concentration 0.15%) and pipetted into 96-well plate containing 200 µL of growth medium (97.5% DMEM/F-12 (Gibco, Waltham, MA, USA), 2.1% NeuroCult including GM supplements, and 0.4% 100 mM penicillin/streptomycin) delivered as a viscous drop. CURC (5 µM) was added to the PM in a dH_2_O mix.

### 2.2. Cell Viability and Apoptosis Assays

iNPC viability was evaluated using the MTS assay (3-(4,5-dimethylthiazol-2-yl)-5-(3-carboxymethoxyphenyl)-2-(4-sulfophenyl)-2H-tetrazolium) following the manufacturer’s instructions (Promega Biotech Ibérica S.L., Madrid, Spain) after one and five days of in vitro culture. Non-embedded iNPCs were cultured in growth medium in the presence of 5 μM of CURC or the corresponding volume of phosphate buffer solution (PBS; as the vehicle) as controls. Optical density was measured at 490 nm using a Victor2 microplate reader (Perkin Elmer Inc., Waltham, MA, USA) 4 h after adding the MTS reagents.

Apoptosis was evaluated using the Apotox-BIO^TM^ Triplex assay kit (Promega, Madison, WI, USA) following the manufacturer’s instructions. Briefly, Caspase-Glo^®^ 3/7 Reagent was added and incubated for 30 min; then, bioluminescence was measured using a Victor2 microplate reader.

### 2.3. Phenotypic Characterization of Embedded iNPCs

PM-embedded iNPCs in the presence or absence of CURC were fixed with 4% paraformaldehyde (PFA) for 15 min, washed three times with PBS, blocked, and then permeabilized with 5% normal goat serum (Thermo Fisher) and 0.1% Triton X-100 (9036-19-5, Merck Millipore, Darmstadt, Germany) for 1 h. The cells were then incubated with primary antibodies overnight at 4 °C in a humidified chamber. The primary antibodies used were Ki-67, proliferation marker (1:400, ab15580, Abcam, Cambridge, UK), Nestin (1:400, ab6142, Abcam), and MAP2 (1:500, ab5392, Abcam), along with Alexa488 and Alexa647 dye-conjugated secondary antibodies. After washing with PBS three times to remove excess primary antibody, samples were incubated for 1 h with dye-conjugated secondary antibodies in a 1:400 dilution in blocking solution (α-mouse Alexa-488, α-rabbit Alexa-647). All samples were counterstained with 4,6-diamidino-2-phenylindole dihydrochloride (DAPI; Invitrogen, Waltham, MA, USA) for 5 min at room temperature. Cell debris and autofluorescent aggregates found either in the scaffolds or the spinal cord not overlapping with the DAPI signal were excluded from the cell quantifications. The samples were examined by confocal microscopy (confocal microscope Leica TCS-SP2-AOBS).

### 2.4. Organotypic Longitudinal Spinal Cord Slice Culture

Spinal cord slices were obtained from five-day-old Sprague–Dawley rats and processed for culture as previously described [14,35]. The spinal cord was dissected and cleaned from the meningeal layers and immersed in ice-cold Hank’s balanced salt solution (HBSS). The spinal cord was then cut using a McIlwain tissue chopper into 350 μm thick slices in the parasagittal longitudinal plane. The slices were then cultured on Millicell cell culture inserts (Millipore) preequilibrated with 1.7 mL of culture medium (50% minimum essential medium, 25% HBSS, 25% horse serum, 2 mM GlutaMAX, 1 mM NAC, 0.5% NaHCO_3_, and 1% penicillin/streptomycin). To mimic an ex vivo model of SCI, a complete transverse section was performed using a scalpel blade on day five. After transection, both PM_iNPC and PM_CURC_iNPC (3 × 10^5^, GFP-overexpressing iNPCs in this case) were seeded above the created gap and cultured for another five days. The samples were then fixed with 4% PFA for 30 min for immunohistochemical analysis.

### 2.5. Preparation of HA Demilune with PLA Fibers for In Vitro Experiments

The synthesis of the demilune HA scaffold as well as the characterization of its mechanical properties was carried out as previously described [17]. Briefly, poly-ε-caprolactone (PolySciences, Warrington, PA, USA; Mw = 40 kDa) fibers of 400 µm were extruded in a Hater Minilab. A thin polytetrafluoroethylene block with 2.5 mm-wide grooves and a single poly-ε-caprolactone fiber of 1 mm diameter were used as a mold to obtain conduits. HA (5% (*w*/*v*)) was dissolved for 24 h in sodium hydroxide 0.2 M (Scharlab, Barcelona, Spain). Then, HA was crosslinked with divinyl sulfone (Sigma-Aldrich; Darmstadt, Germany; 10:9 monomeric unit molar ratio), mixed, injected into a mold, and then lyophilized for 24 h (Lyoquest-85, Telstar, Bensalem, PA, USA) to generate microporous HA cylinders. Longitudinal cuts generated the demilune scaffolds. Finally, the HA demilune scaffolds were hydrated in distilled water for 2 h, the poly-ε-caprolactone fibers extracted, and the conduits cut to the desired length of 4 mm.

### 2.6. Preparation, Combination with HA Demilune Scaffolds, and Sanitization of PPY-Coated Microfibers

Highly aligned microfiber bundles (AITEX Textile Research Institute, Alcoy, Spain) formed by 300 PLA microfibers with a diameter of 10 µm were coated with PPY via in situ polymerization [30,36]. Briefly, the microfibers were introduced into a polypropylene tube containing an aqueous solution of 14 mM pyrrole monomer (Py, Sigma-Aldrich 131709) and 14 mM sodium para-toluene sulfonate (pTS, Sigma-Aldrich, 152536). Ultrasonication for 1 min allowed the microfibers to become saturated with the Py/pTS solution. The microfibers were then incubated with shaking at 4 °C for 1 h. Next, an aqueous solution of 38 mM ferric chloride (Sigma-Aldrich 157740) was added and incubated with shaking at 4 °C for 24 h for the polymerization and deposition of PPY onto the PLA microfibers. The PPY-coated microfibers were washed three times with deionized water with agitation for 10 min and then ultrasonicated for 30 min in deionized water three times. Finally, the microfibers were dried in a desiccator with a fixed vacuum at 40 °C for 48 h. Once the PPY coating was completed, one microfiber bundle was introduced into the lumen of the HA scaffold. HA droplets were added at both ends of the PPY-coated microfiber bundle to fix the HA demilune scaffold. In the following text, ‘PPY fibers’ is used as shorthand for ‘PPY-coated PLA fibers’ for the sake of brevity. Before cell seeding, the HA demilune scaffolds with PPY-coated fibers were sanitized for 2 h with 70% ethanol (Scharlab) and then 10 min with 50% ethanol, 30% ethanol, and then distilled water. The conduits were incubated with culture medium overnight.

### 2.7. Scanning Electron Microscopy

The surface morphology of the biomaterials was characterized using scanning electron microscopy (SEM; ULTRA 55, ZEISS Oxford Instruments, Pleasanton, CA, USA). The samples were desiccated under vacuum conditions 24 h prior to evaluation to avoid interference due to evaporated water. The samples were placed on carbon tape, and a carbon bridge was introduced between the sample and the carbon tape. Finally, the samples were coated with a thin layer of platinum, and the images were taken using a voltage of 2 kV.

SEM was also used to evaluate the attachment/morphology of PM-embedded iNPCs seeded within HA and HA-PPY conduit scaffolds after three days of culture. This mode of SEM was performed as described previously [17]. Briefly, tissues were placed into pre-warmed 3.5% (*v*/*v*) glutaraldehyde (Sigma) and 2.5% (*v*/*v*) PFA (Sigma) and then left to incubate overnight at 4 °C. The constructs were then coated with a conductive layer using 2% *w*/*v* osmium tetroxide for 1 h. This step was followed by dehydration using a serial dilution of ethanol (30%, 50%, 70%, 96%, and 100%) and hexamethyldisilane treatment. The scaffolds were then air-dried at room temperature overnight. Images from different spots were examined under a transmission electron microscope (FEI Tecnai G2 Spirit BioTwin, Thermo Fisher Scientific company, Waltham, Massachusetts, USA) using a Morada digital camera (EMSIS GmbH, Münster, Germany).

### 2.8. In Vivo SCI Model and Biomaterial Implantation

SCI was induced in female Sprague–Dawley rats (weighing ~200 gm) as previously described [37]. Briefly, the animals were treated with subcutaneous morphine (2.5 mg/kg) 30 min before surgery and then deep anesthetized with 3% isoflurane, which was maintained at 1.5–2% during surgery for SCI induction. Laminectomies were conducted at thoracic segments T7–T9 for a moderate contusion at T8 by applying 200 kdyn in all animals using the Infinite Horizon spinal cord impactor (Precision Systems and Instrumentation LLC, Brimstone, VA, USA). One week after SCI, animals were randomly distributed into the following groups (*n* = 3) (See Table 1 for details): HA_PM_CURC, HA_PM_CURC_iNPC, HA_PM_iNPC, HA_PPY_PM_CURC, HA_PPY_PM_CURC_iNPC, and HA_PPY_PM_iNPC. The iNPCs (1 × 10^6^) were embedded in 10 µL of PM (0.15%). The SCI area was reassessed, a slit was made in the dura mater with a 27G needle, and the biomaterial was placed over the spinal cord, as previously described [17]. A laminectomy was performed without applying the injury in the control groups: HA_PPY_PM_iNPC (non-injured) and HA_PM_iNPC (non-injured). After surgery, all animals were subjected to post-operative care. To prevent immune rejection of allogeneic cell grafts, animals received daily subcutaneous injections of the immunosuppressant tacrolimus (1 mg/kg) starting one day before transplantation, until sacrifice one week after implantation.

### 2.9. Histological Studies

One week after scaffold implantation and SCI, animals were irreversibly anesthetized by intraperitoneal injection of sodium pentobarbital (100 mg/kg) and fentanyl (0.05 mg/kg) and transcardially perfused with 0.9% saline immediately followed by 4% PFA in 0.1 M phosphate buffer (pH 7.4). Spinal cords were dissected and post-fixed in 4% PFA for 5 h and then conserved in 0.1 M phosphate buffer containing 0.01% sodium azide. Thoracic segments, including T6 to T10, were dehydrated and included in paraffin, placed in histology cassettes, and processed on a Leica ASP 300 tissue processor (Leica Microsystems, Nussloch, Germany). Then, 8 mm thick sagittal sections were cut on a microtome and mounted on gelatin-coated slides. For histological analysis, spinal cord tissue sagittal sections were permeabilized with PBS containing 0.5% Triton and 2% goat serum (blocking solution). The primary antibodies used were β-Tubulin III (1:400, MO15052 Neuromics, Edina, MN, USA), GFAP (1:500, Z0334 DAKO, Santa Clara, CA, USA), GFP (1:750, ab13970, Abcam), ED1 CD68 (1:400, MAB1435 Chemicon-Fisher Scientific, Madrid, Spain), Iba-1 (1:400, 019-19741 DAKO, Santa Clara, CA, USA), NeuN (1:600, ABN91 Sigma-Aldrich), Fibronectin (1:50, SC6953, Santa Cruz, USA), Neurofilament (1:750, ab24575, Abcam), and PDGF (1:400, ab32570, Abcam). AlexaFluor-488, -555, or -647 (1:400 Invitrogen) conjugated with secondary antibodies were used. All sections were counterstained with DAPI and mounted using FluorSave TM Reagent (EMD, Millipore). Mounted sections were scanned by an Aperio Versa scanner (Leica Biosystems, Germany) and analyzed using the ImageJ (Bethesda, MD, USA).

### 2.10. Ethical Statement

All experimental procedures were approved by the Animal Care and Use Committee of the Research Institute Prince Felipe (2021/VSC/PEA/0032) in accordance with the guidelines established by the European Communities Council Directive (210/63/EU) and Spanish regulation 1201/2005.

### 2.11. Statistical Analysis

In vitro experimental data were collected from three independent experiments, and the results were reported as the mean ± standard deviation (SD). For the comparisons between two groups of values, the statistical analysis of the results was performed using the student’s *t*-test for normally distributed data. Two-way ANOVA with appropriate corrections (such as Tukey’s post hoc test) was used to compare groups in viability, apoptosis, and MTS assays. To compare immunofluorescent expression in histological specimens in distinct groups, one-way ANOVA was used; however, the comparison of surface marker expression of Nestin, Ki-67, and MAP2 between PM_iNPC and PM_CURC_iNPC in vitro used a paired t-test. Statistical analyses were performed using GraphPad software (La Jolla, CA, USA). Differences were considered significant at * *p* < 0.05, ** *p* < 0.01, *** *p* < 0.001, and **** *p* < 0.0001.

## 3. Results

### 3.1. PuraMatrix-Embedded iNPCs Increase Long-Term Survival

To evaluate the viability of iNPCs embedded in the PM hydrogel and 3D in vitro culture in the presence or absence of CURC, we first performed an MTS assay to evaluate cell metabolic activity after one and five days of culture (Figure 1A). Interestingly, embedding cells in PM in the presence or absence of CURC (PM_CURC_iNPC and PM_iNPC, respectively) significantly reduced the metabolic activity of iNPCs during the first day of culture compared with iNPCs cultured under traditional conditions (iNPC or CURC_iNPC). However, we found the opposite profile at day five, with increased metabolic activity in PM-embedded iNPCs compared to traditional 2D iNPC cultures (Figure 1A). Overall, this suggests that the 3D culture conditions in PM support the long-term viability of iNPCs.

In this 2D/3D comparison, we also evaluated cell death/apoptosis using the Apotox-BIO^TM^ Triplex assay, which employs a pro-luminescent caspase-3/7 substrate (Figure 1B). The results indicated a substantial reduction in cell apoptosis in PM-embedded iNPCs in the presence of CURC (PM_CURC_iNPC) compared to non-embedded iNPCs. Overall, these findings suggest that PM and CURC act synergistically to reduce iNPC apoptosis in long-term culture.

### 3.2. CURC Induces a Neurogenic-like Phenotype of PM-Embedded iNPCs

We next evaluated the neurogenic phenotype of PM-embedded iNPCs by assessing cells with an elongated morphology, the expression of the neural stem cell intermediate filament protein Nestin, and cell proliferation (via Ki-67 staining) (Figure 2A). Overall, PM-embedded iNPCs in the presence of CURC (PM_CURC_iNPC) displayed a 2.2-fold greater proportion of elongated Nestin-positive cells (red) than in the absence of CURC (PM_iNPC). However, we failed to encounter any significant differences between PM_iNPC and PM_CURC_iNPC in terms of cell proliferation (Ki-67—green staining) (Figure 2B). Interestingly, we discovered that CURC increased the number of MAP2-positive cells in 3D culture, indicating a potential effect on neuronal cell fate maturation (Appendix A).

We next cocultured GFP-labeled iNPCs with organotypic spinal cord slices to mimic the process of cell transplantation in an ex vivo model of SCI. As shown in Figure 3A, we transversally transected longitudinal slices of neonatal spinal cords and deposited PM-embedded iNPCs in the presence or absence of CURC (PM_CURC_GFP_iNPC and PM_CURC_iNPC) into the gap. After five days of culture, and in agreement with our previous findings, we found that CURC promoted iNPCs to take on a neuron-like morphology. The iNPCs also displayed long neurite-like processes with a parallel orientation in tight association with endogenous neuronal fibers stained with βIII-tubulin (Figure 3B, red, white arrows). In the absence of CURC, iNPCs took on a wider morphology and a random organization (Figure 3B, yellow arrow). The quantification of the GFP-iNPCs with neurite-like processes demonstrated a significantly increased number following treatment with CURC (Figure 3C).

### 3.3. HA Demilune Scaffolds Containing PPY-Coated Fibers Guide PM-Embedded iNPCs for Spinal Cord Implantation

As shown in Figure 4A, we obtained HA demilune scaffolds with the proper dimensions for implantation within the lesioned rat spinal cord. The lumen of the HA scaffold contained a bundle of highly aligned PPY microfibers. Figure 4B provides evidence of the high porosity of the HA scaffold, while Figure 4C demonstrates the parallel alignment of the PPY microfibers. Figure 4D shows the surface morphology of the PM adsorbed onto the fiber surface, while Figure 4E depicts the grainy topography of the PPY fibers.

We performed additional SEM analyses to assess construct microarchitecture and evaluate the cell distribution within the HA scaffold and PPY fibers. Figure 4F provides a schematic diagram of scaffolds (HA alone and HA with PPY fibers) seeded with PM-loaded iNPCs. The SEM images in Figure 4G,H depict iNPCs embedded within the PM hydrogel covering the HA scaffold internal surface (Figure 4I, white arrow) and iNPCs attached to PPY fibers (Figure 4J, black arrow).

### 3.4. CURC Supports iNPC Migration within the Injured Spinal Cord

We implanted PPY fiber-functionalized HA scaffolds carrying PM-embedded GFP-iNPCs in the presence or absence of CURC to cover the damaged segments of the spinal cord one week after injury, mimicking the most favorable clinical intervention at the sub-acute phase (Figure 5A). To evaluate the damage induced by scaffold implantation in healthy tissue, we also implanted the various scaffolds in non-injured animals and evaluated their impact after one week. We stained sagittal sections with hematoxylin and eosin (H&E) to analyze macroscopic anatomical alterations. An overall view of the spinal cord revealed the formation of a fibrous-like tissue pad under the implantation site in all cases (Figure 5B, white arrow); however, we failed to observe any structural pathological alterations (e.g., cysts or infiltrated cells) at this stage in the injured groups (Figure 5B).

To evaluate the migration of GFP-iNPCs after implantation, we tracked GFP-positive cells within the remaining implanted biomaterial (Figure 6A, left panels) and in the spinal cord tissue (Figure 6A, right panels). One week after injury and implantation, we detected most GFP-iNPCs within the remaining biomaterial (Figure 6B, upper panel). While the global iNPC survival rate was lower in the implanted injured groups compared with non-injured group (data not shown), we hypothesized that the cells that migrated into the injured samples died faster than those migrating into the non-injured tissue and, in addition, the cells remaining in the scaffold were able to survive for a longer period. Nevertheless, we discovered an inverse correlation between the percentage of GFP-positive iNPCs in the biomaterial vs. GFP-iNPCs detected in the spinal cord tissue. It is of note that the presence of CURC prompted iNPC migration from the biomaterial to the spinal cord tissue (Figure 6B,C, black), while the presence of PPY delayed migration (Figure 6B,C, light blue).

We also observed that the few GFP-iNPCs that integrated into the host tissue displayed co-localization with βIII-tubulin expression in all of the groups, indicating a general lack of neuronal differentiation after implantation (Figure 6D); however, most GFP-positive iNPCs co-localized with GFAP staining in all evaluated conditions, which indicates preferential astroglial differentiation (Figure 6E).

### 3.5. A Fully Functionalized HA Demilune Scaffold Preserves Neuronal Fibers and Reduces Glial Scarring but Does Not Reduce Neuroinflammation Early after SCI

Assessments of neuronal preservation after SCI by evaluating the expression of neuronal filaments with βIII-tubulin staining (Figure 7A, red) demonstrated that only the fully functionalized scaffold (PM-embedded GFP-iNPCs with PPY fibers and CURC) allowed for the preservation of β-III-tubulin-positive neuronal fibers after implantation (Figure 7B, light blue), as all other approaches led to a significant decrease in fiber preservation compared to the control. Analysis of scar-forming tissue (as delimited by the GFAP-negative area; Figure 7A, white dash line) revealed that the group of animals implanted with the fully functionalized scaffold showed a smaller scar area, in agreement with the above-described neuronal preservation effect (Figure 7C, light blue).

We next evaluated both endogenous microglial activation using Iba1 immunoreactivity, describing activated and resting microglia as displaying a rounded or stellated morphology, respectively (Figure 8A) [38], and macrophage infiltration by ED1 immune reactivity (Figure 8B) [39]. Quantification demonstrated a significant reduction in resting microglia and increased activated microglia in all injured groups compared to the non-injured group, with no significant difference observed between groups (Figure 8C). The analysis of infiltrated macrophages revealed a similar result; we failed to find significant differences between the treatment approaches (Figure 8). Overall, these data suggest that the implants did not show any significant immunomodulatory effect one week after implantation.

Finally, we evaluated the fibrotic formed tissue at the injury site by evaluating the expression of platelet-derived growth factor (PDGF), a pericyte marker involved in fibrotic scar tissue generation [40]. We found that the fully functionalized scaffold induced the lowest PDGF expression (Figure 9A,B). Nonetheless, we failed to find differences in the spared neuronal area among the different injured groups (as confirmed by NeuN expression), indicating a lack of early spinal neuronal preservation after scaffold implantation (Figure 9C,D).

## 4. Discussion

In the present study, we report that iNPCs embedded into a PM hydrogel at a concentration of 0.15% containing 5 µM of CURC display improved cell survival compared to a long-term monolayer culture. CURC within the PM hydrogel prompts a higher number of neuronal-like cells with elongated projections. When implanted in the injured rat spinal cord (capping the injured area), the combination of PM-embedded iNPCs with CURC prompted the preservation of neuronal fibers.

The pH-dependent biodegradable polyacetal’s functionalization by the conjugation with curcumin improved its properties in comparison with the free form, including increased water solubility and stability, reduced cell toxicity, and enhanced pH-mediated controlled release in the acidic pH of the lesion site as we previously described [14]. We previously demonstrated the neuroprotective effects of CURC treatment in vitro and in vivo [14,41]. In 2D iNPC culture, CURC treatment induced higher preservation of neuronal-like iNPCs positive for β-III-tubulin when exposed to toxic doses of H_2_O_2_. CURC treatment alone or in combination with NPC transplantation in a rat model of chronic stage SCI improved neuronal fiber preservation within the injured area. In the present study, and in agreement with our previous results, we found that the fully functionalized scaffold containing CURC markedly preserved β-III-tubulin neuronal fibers, evaluated at an earlier stage of one week after implantation and two weeks after injury.

Previous studies had suggested that PPY stimulated the release of neurotransmitters and trophic factors [42], although our previous in vitro studies underscored that PPY fibers failed to alter brain-derived neurotrophic factor secretion from Schwann cells [36]. Nevertheless, while only the fully functionalized scaffold that included PPY fibers demonstrated significant neuronal fiber preservation, we hypothesize that PPY fibers may synergize with CURC within the iNPC transplant. Overall, both embedded iNPCs that remained in the hydrogel and those that migrated to the spinal cord parenchyma could enhance endogenous trophic support. We previously demonstrated that CURC reduced the glial scar when intrathecally administered [14], possibly by reducing astrogliosis and glial scar formation by inhibiting the signal transducer and activator of transcription 3 (STAT3) and the nuclear factor-kappa B (NF-κB) pathways [43,44]. Interestingly, the fully functionalized scaffold also contributed to reducing the glial scar.

We also previously demonstrated that CURC reduced macrophage infiltration [41] and promoted the anti-inflammatory polarization of microglia [14] at two months after injury/treatment. This study failed to find a difference in microglial activation or macrophage infiltration one week after treatment (implantation). Further experimentation evaluating microglia activation at chronic stages will be required to fully assess the potential anti-inflammatory effect of the CURC containing scaffolds. A study on traumatic brain injury revealed that the transplantation of PM-embedded rat NPCs alone or in combination with free curcumin decreased microglial activation and macrophage infiltration when evaluated at one month after treatment [45,46].

We found that the scaffolds containing CURC induced a lower level of fibrosis between the scaffold and the spinal cord tissue, as evaluated by a macroscopic analysis of the H&E staining (data not shown). A recent study developed in a laminectomy model in which curcumin prevented peridural fibrotic tissue formation supports this finding [47]. The anti-fibrotic properties could be attributed to the capability of curcumin to suppress collagen production through down-regulation of transforming growth factor-beta 1 [48,49]. Additionally, pericytes have been determined as a target to reduce fibrotic scarring and improve tissue recovery [50,51], a finding that has prompted the development of PDGF inhibitors as a therapeutic approach to systemic sclerosis [52]. We monitored the expression of PDGF to evaluate the potential modulation of the fibrotic tissue within the SCI by the different scaffolds. We discovered that a fully functionalized scaffold reduced scar formation and increased neuronal preservation, supporting neuronal regeneration.

Curcumin also enhances neural differentiation of pluripotent embryonic carcinoma cells and induced NeuroD, TUJ1, and PAX6 expression through the Notch signaling pathway [53]. In vitro, we found more iNPCs expressing MAP2, a mature neuronal marker, in the presence of CURC (Appendix A); however, very few implanted cells showed reactivity to βIII-tubulin in vivo, and we failed to observe significant differences induced by CURC. Nevertheless, Woodruff and collaborators Lu, et al. [54] reported that transplanted iNPCs expressed β-III-tubulin after three months post-transplantation, although these cells rarely displayed mature neuronal markers. Exploring the neuronal phenotype of iNPCs one week after implantation may be too early to induce neuronal in vivo differentiation. Interestingly, most GFP-iNPCs located surrounding the injured area displayed positive staining for GFAP, indicating a preferential glial fate rather than neuronal differentiation at this stage.

The HA demilune scaffolds containing a bundle of highly aligned PPY fibers in their lumen increased neural cell adhesion up to 95% and favored cell alignment and elongation [30,55]. The PPY coating of the PLA fibers confers a surface nanoroughness that enhances the adsorption of extracellular matrix proteins and thus makes these surfaces more adherent [30,36]. This mechanism may explain the low migration of transplanted cells to the spinal cord, which remains more firmly attached to the fibers. The rigidity of the PPY fibers could also contribute to the lower performance observed when employing non-pre-seeded microfibers. Reducing the number of implanted microfibers and/or using a less rigid support material like silk fibroin may improve this result.

CURC enhanced iNPC migration and homing into spinal cord tissue, which agrees with a previous study [56] in which CURC enhances cell migration through c-Src and protein kinase C phosphorylation and activation. Nevertheless, when CURC-containing scaffolds included PPY, we found no significant differences in iNPC migration. Therefore, the adhesion of iNPCs to PPY fibers may inhibit CURC-stimulated migration.

## 5. Conclusions

A complementary combination of CURC and iNPCs supported by the HA demilune scaffold containing PPY fibers prevented neuronal degeneration soon after severe SCI by preserving neuronal fibers and reducing glial scar formation without significantly additionally damaging the preserved tissue. Nevertheless, further studies are needed to evaluate the consequences for long-term transplantation of the scaffolds employed in this study.

## Figures and Tables

**Figure 1 biomedicines-09-01928-f001:**
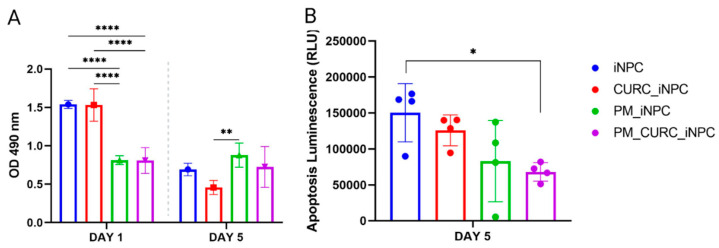
Evaluation of iNPC viability and apoptosis. (**A**) Cell metabolic activity/cell viability evaluated by MTS assay and (**B**) apoptosis evaluated using the Apotox-BIO^TM^ Triplex assay of iNPCs in vitro cultured in traditional 2D or PM-embedded (0.15%) 3D conditions in the presence or absence of CURC (5 µM) at one and five days. Data represented as the mean ± SD (* *p* < 0.05, ** *p* < 0.01, and **** *p* < 0.0001).

**Figure 2 biomedicines-09-01928-f002:**
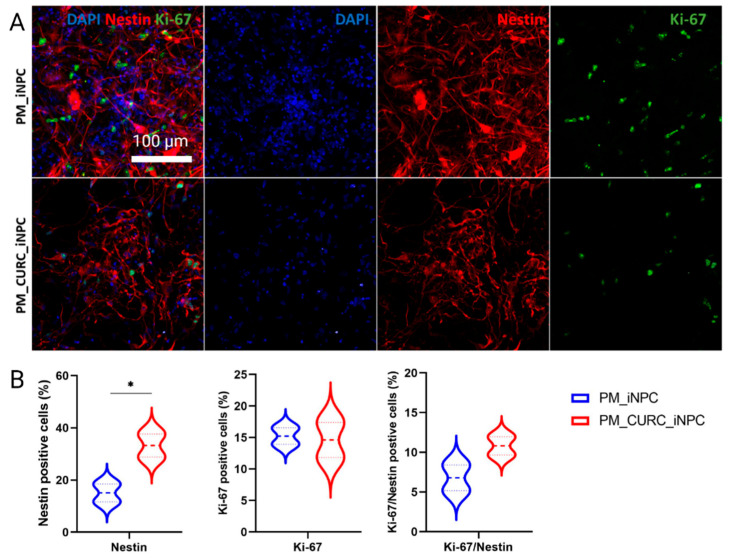
Quantitative and qualitative analysis of the neurogenic-like phenotype in iNPCs in 3D culture. (**A**) Confocal immunofluorescence images for Nestin (red) and Ki-67 (green) in PM-embedded iNPCs in the presence or absence of 5 µM CURC. Nuclei stained with DAPI (blue). White scale bar: 100 µm (all images were acquired at the same magnification). (**B**) Quantitative analysis of Nestin and Ki-67expression in iNPCs. Data represented as the mean ± SD from three independent experiments (* *p* < 0.05).

**Figure 3 biomedicines-09-01928-f003:**
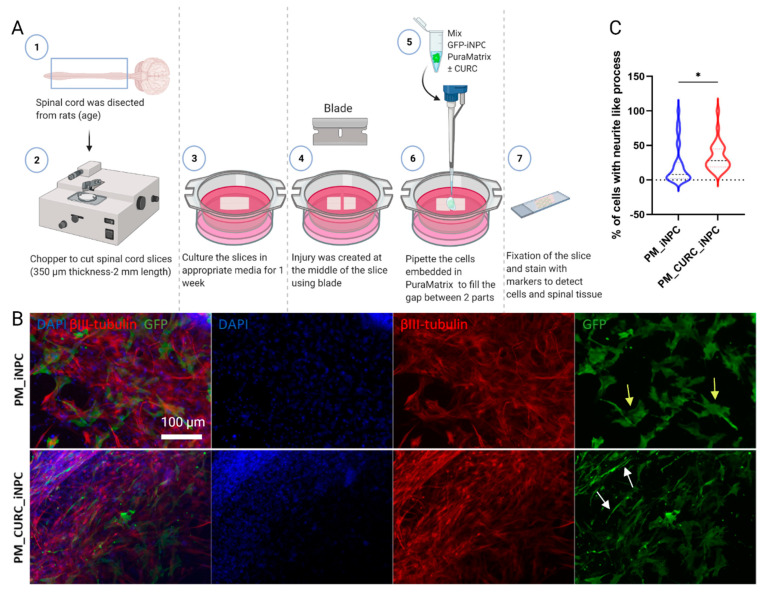
Assessment of iNPC morphology when cocultured with organotypic spinal cord slices. (**A**) Schematic diagram of the experiment. (**B**) Representative immunofluorescent images showing spinal cord fixed tissue labeled with βIII-tubulin (red) and GFP-labelled iNPCs (green). Nuclei stained with DAPI (blue), white scale bar: 100 µm (all images were acquired at the same magnification). (**C**) Quantitative analysis of neurite outgrowth elongation from iNPCs. Data represented as mean ± SD from three independent experiments (* *p* < 0.05).

**Figure 4 biomedicines-09-01928-f004:**
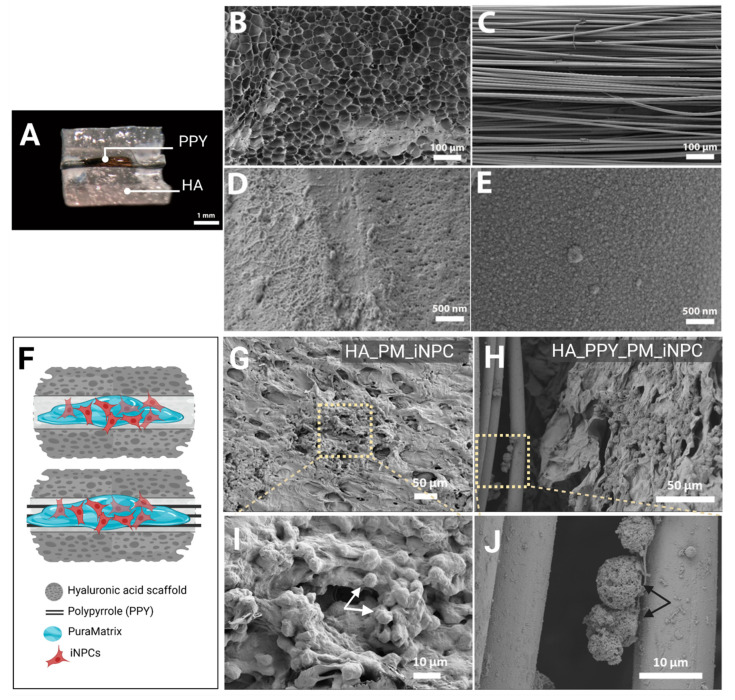
Morphological analysis of iNPCs and scaffold components. (**A**) Macroscopic image of the HA demilune scaffold containing PPY microfibers in the lumen (hydrated state), scale bar: 1 mm. (**B**) SEM image detailing HA scaffold porosity. (**C**) SEM image showing the parallel alignment of PPY microfibers. (**B**,**C**) scale bar: 100 µm. (**D**) SEM image of PM on the PPY microfiber surface. (**E**) SEM image showing the PPY coating of the PLA microfiber surface. (**B**,**C**) scale bar: 500 nm. (**F**) Schematic diagram of scaffolds (HA alone and HA with PPY fibers) seeded with PM-embedded iNPCs. (**G**,**H**) SEM images of PM-embedded iNPCs seeded on HA (**G**) and HA-PPY (**H**) after three days in culture, scale bar: 50 µm. (**I**,**J**) show a higher magnification of the indicated area in (**G**,**H**) respectively, scale bar: 10 µm.

**Figure 5 biomedicines-09-01928-f005:**
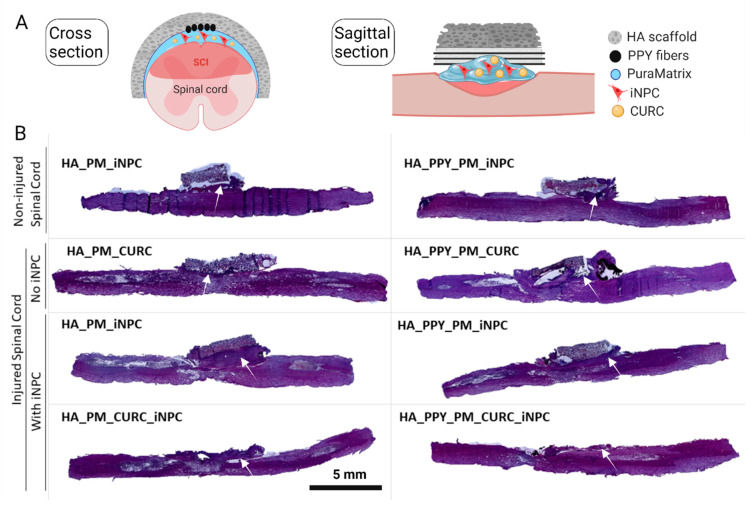
Implantation of demilune scaffolds to cap the SCI. (**A**) Schematic diagram showing the cross and sagittal sections of the SCI and various scaffolds capping the injured area. (**B**) Representative images of H&E staining of sagittal spinal cord sections from one animal per group one week after injury. The white arrow indicates the formation of a fibrous-like tissue pad. Scale bar: 5 mm.

**Figure 6 biomedicines-09-01928-f006:**
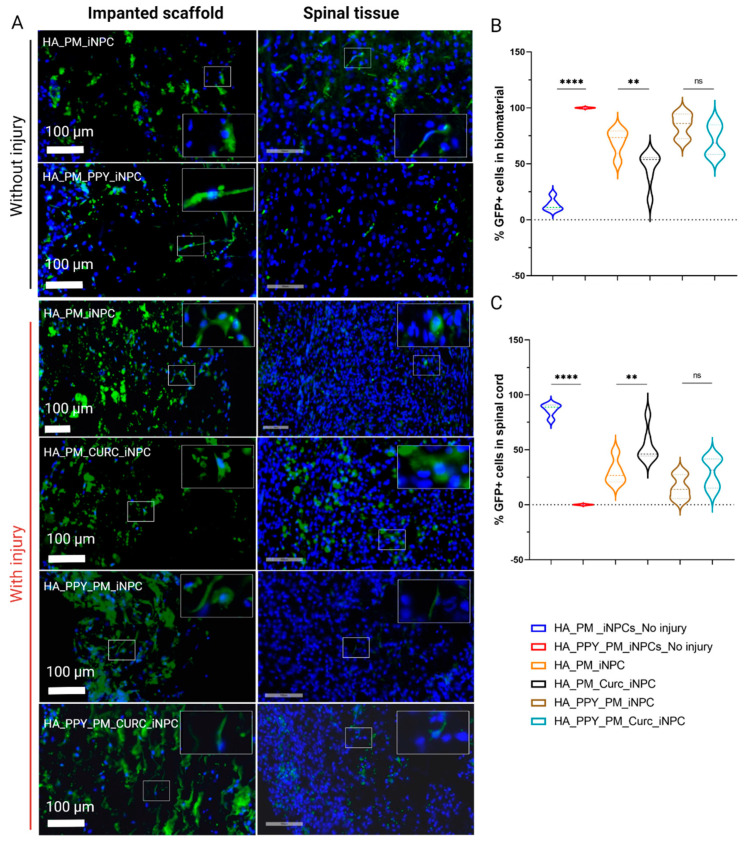
Quantitative and qualitative analysis of iNPC biodistribution and differentiation potential after implantation into the spinal cord. (**A**) Representative images showing the distribution of GFP-iNPCs (green) within the biomaterial scaffold and spinal cord tissue. Cell nuclei stained with DAPI (blue). Scale bar: 100 µm. (**B**) Quantitative analysis of GFP-iNPCs in the biomaterial scaffold and (**C**) in the spinal cord. Data represented as mean ± SD, *n* = 3 (^ns^
*p* > 0.05, ** *p* < 0.01, **** *p* < 0.00001); representative images of GFP-iNPCs (green) expressing (**D**) β-III-tubulin (red) and (**E**) GFAP (orange). Scale bar: 50 µm.

**Figure 7 biomedicines-09-01928-f007:**
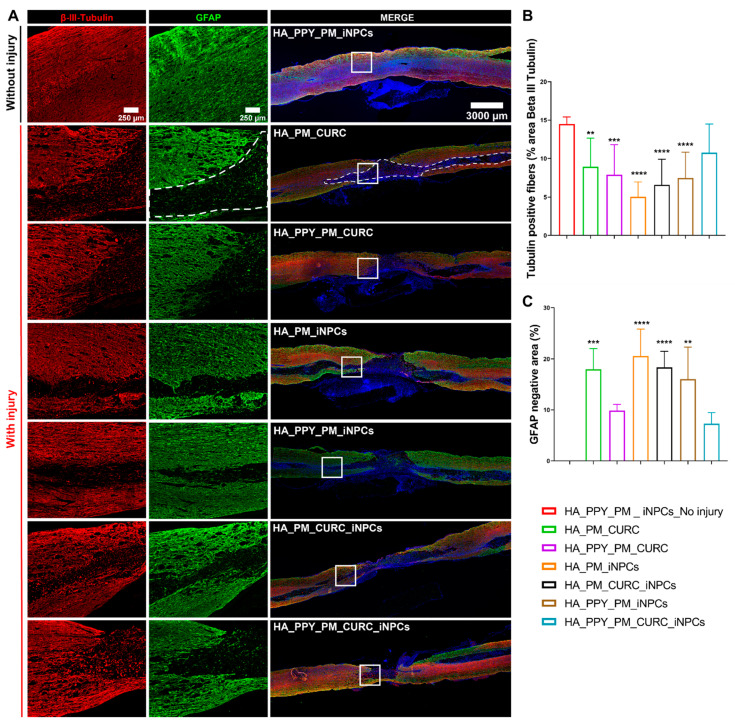
Potential of biomaterial and iNPCs to preserve neuronal fibers and reduce scar formation within the SCI. (**A**) Immunofluorescent staining of β-III-tubulin (neuronal marker, red) and GFAP (astroglial marker, green). (**B**) Quantification of β-III-tubulin expression. (**C**) GFAP negative area. Data represented as means ± SD *n* = 3, (** *p* < 0.01, *** *p* < 0.001, **** *p* < 0.0001 vs. PM_PPY_iNPC_no injury). Scale bars: 250 µm; 3000 µm.

**Figure 8 biomedicines-09-01928-f008:**
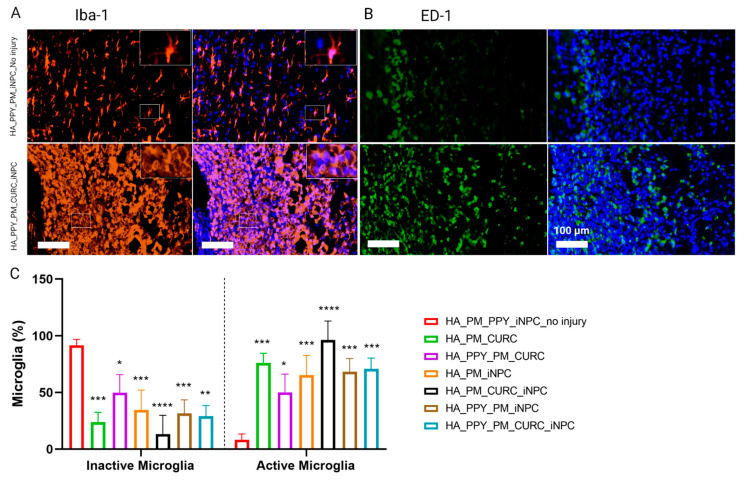
Evaluation of the neuroinflammatory reaction at the early stages after SCI. (**A**) Immunofluorescent staining for Iba-1 (microglial marker, orange) and (**B**) ED-1 (macrophage marker, green). Scale bar: 100 µm. (**C**) Quantification of inactive and active microglia identified by the morphological appearance of Iba-1 expressing cells. Data represented as mean ± SD, *n* = 3 (* *p* < 0.05, ** *p* < 0.01, *** *p* < 0.001, **** *p* < 0.00001).

**Figure 9 biomedicines-09-01928-f009:**
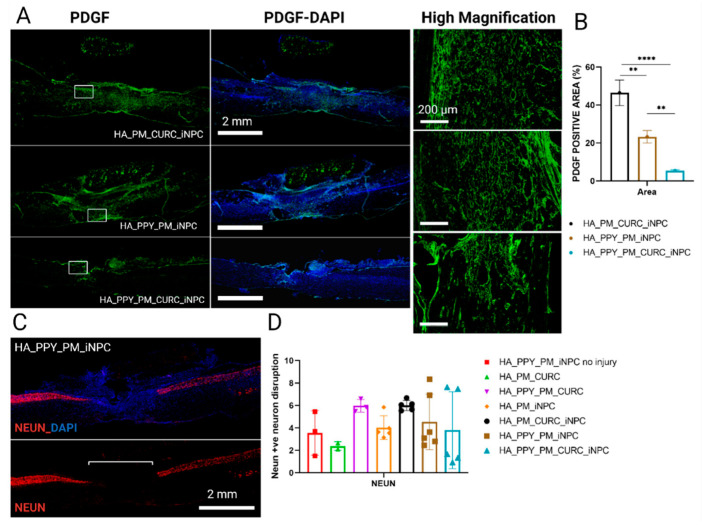
Quantitative and qualitative analysis of PDGF and NeuN expression at the site of injury. (**A**) Immunofluorescent staining for PDGF (green) at low magnification (left panels, including merged images with DAPI staining), Scale bar: 2 mm and a selected area at high magnification (right panel), Scale bar: 200 µm. (**B**) Quantification of PDGF positive expression area. Data represented as mean ± SD by one-way ANOVA with Tukey’s multiple comparison test (** *p* < 0.01, **** *p* < 0.0001). (**C**) Representative image of NeuN (red) and DAPI staining. Scale bar: 2 mm. (**D**) Quantification of the disruption of the NeuN-positive neuron area at the injury site. Data represented as mean ± SD by one-way ANOVA with Tukey’s multiple comparison test, *n* = 3.

**Table 1 biomedicines-09-01928-t001:** Experimental groups.

Groups	Induction of SCI	Treatment
HA_PM_iNPC (non-injured)	No	Hyaluronic acid scaffold with PM-embedded iNPCs
HA_PPY_PM_iNPC (non-injured)	No	Hyaluronic acid scaffold with PPY fibers and PM-embedded iNPCs
HA_PM_CURC	Yes	Hyaluronic acid scaffold and PM-embedded CURC
HA_PPY_PM_CURC	Yes	Hyaluronic acid scaffold with PPY fibers and PM-embedded CURC
HA_PM_iNPC	Yes	Hyaluronic acid scaffold with PM-embedded iNPCs
HA_PPY_PM_iNPC	Yes	Hyaluronic acid scaffold with PPY fibers and PM-embedded iNPCs
HA_PM_CURC_iNPC	Yes	Hyaluronic acid scaffold with PM-embedded iNPCs and CURC
HA_PPY_PM_CURC_iNPC	Yes	Hyaluronic acid scaffold with PPY fibers, PM-embedded iNPCs, and CURC

Abbreviations—HA: hyaluronic acid; PM: PuraMatrix; PPY: polypyrrole fibers; iNPCs: human-induced neural progenitor cells; CURC: nanoconjugated curcumin.

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
