# Peer review of "A Hyaluronic Acid Demilune Scaffold and Polypyrrole-Coated Fibers Carrying Embedded Human Neural Precursor Cells and Curcumin for Surface Capping of Spinal Cord Injuries"

_biomedicines, 2021, doi:10.3390/biomedicines9121928_

Round 1

Reviewer 1 Report

Very well designed and executed study. It was interesting that the PPY microfibers in the biocomposite did not perform as one might have expected.

Author Response

We really appreciate the very favourable reviewer´s comments.

Reviewer 2 Report

In this manuscript, the authors report on biomaterials, which include HA scaffold, PPY fibers, iNPCs, and curcumin, for the treatment of spinal cord injury (SCI). Overall, the authors present a potential approach for SCI. There are a number of points that should be addressed to improve the manuscript.

  1. Instead of immediate transplantation after SCI, the authors claim that the treatment starts from one-week post SCI (Line 266). The rationale should be described.
  2. The authors claim that CURC can enhance iNPC survival. However, compared with no CURC groups, the viability of iNPC with CURC decreased after 5 days of incubation (Figure 1A). How do the authors explain that why CURC reduces apoptosis of iNPC but the readout of MTS assay, which demonstrates both cell number and cell viability, is not increased?
  3. The authors find that the PM-embedded iNPC reduces metabolic activity. In addition, there is no increase in the OD490nm after 5 days incubation, indicating that there’s almost no cell proliferation from day 1 to day 5. It’s hard to imagine that PM would reduce cell viability and inhibit cell proliferation. How do the authors explain that?
  4. The sample size of Figure 2B, 3C, 6B, 7B, 7C, 8C, 9B, and 9D should be clarified.
  5. The scale bar in Figure 5B is missing.
  6. Figure6 B and C illustrate the same information. Instead of the percentage of the cell, cell counts in the biomaterials and the spinal cord can demonstrate more information of iNPC survival and proliferation besides its distribution.
  7. As illustrated in Figure 6C, almost all of the cells migrate to the spinal cord in the HA_PM_iNPCs group without injury. But in Figure 6A, there are many cells in the scaffold. How do the authors explain the conflicts between fluorescence imaging and quantification? In addition, cell migration into the spinal cord is significantly decreased in the HA_PM_iNPC group with injuries. Why does the migration decrease with injury signals?
  8. In the figure caption of Fig.7A, GFAP is green, not orange.
  9. It would be better to show the whole injury area instead of high magnification images in Figure 9A.
  10. In Figure 9C, it would be great to have images that demonstrate Neun expression for other treatment groups.
  11. The guidelines of abstracts (Line 41-47) could be removed.
  12. The sentence in Line 119-120 is confusing. It would be great if the authors could clarify which condition, with or without PPY, could help axonal growth?

Author Response

We appreciate the reviewer´s comments and constructive criticism of the manuscript data, which improve and accurate the quality of our conclusions. We have followed an attend the vast majority of reviewer´s requests in order to improve the figures and better clarify the significance of our findings, by answering to the specific and detailed comments found below.

  1. Instead of immediate transplantation after SCI, the authors claim that the treatment starts from one-week post SCI (Line 266). The rationale should be described.

We have implanted the scaffold one week after the injury, in the subacute phase of the lesion, mimicking the most probable clinical intervention. Now we have included this rationale in lane 407:

“We implanted PPY fiber-functionalized HA scaffolds carrying PM-embedded GFP-iNPCs in the presence/absence of CURC to cover the damaged segments of the spinal cord one week after injury, mimicking the most favorable clinical intervention at the sub-acute phase” 

  1. The authors claim that CURC can enhance iNPC survival. However, compared with no CURC groups, the viability of iNPC with CURC decreased after 5 days of incubation (Figure 1A). How do the authors explain that why CURC reduces apoptosis of iNPC but the readout of MTS assay, which demonstrates both cell number and cell viability, is not increased?

The reviewer´s comment is correct regarding our conclusion on CURC in long survival effect into the PM since the viability data measured by MTS as metabolic activity is not significantly different compared to rest of conditions. However, since we found that the apoptotic levels were significantly reduced in the presence of CURC, we estimated that the viability/apoptosis rate was in favor of a cell survival promoted by the presence of CURC. However, we have now modified our conclusions and described Figure 1 according to the obtained data, highlighting the effect of PM itself increasing cell metabolism and significantly reducing the apoptosis when CURC is present. We apologize for the mistake and we thanks to reviewer´s comment which will improve and better defined the obtained results.

  1. The authors find that the PM-embedded iNPC reduces metabolic activity. In addition, there is no increase in the OD490nm after 5 days incubation, indicating that there’s almost no cell proliferation from day 1 to day 5. It’s hard to imagine that PM would reduce cell viability and inhibit cell proliferation. How do the authors explain that?

Previous results have shown that encapsulation of cells within the PM significantly reduced proliferation rate compared to surface plating or injection method [Aligholi H et al, 2016]. Our described protocol, however, showed that PM encapsulation maintains the cell survival rate without a drastic reduction effect on proliferation.

Aligholi, H.; Rezayat, S.M.; Azari, H.; Ejtemaei Mehr, S.; Akbari, M., et al., Preparing neural stem/progenitor cells in PuraMatrix hydrogel for transplantation after brain injury in rats: A comparative methodological study. Brain Research, 2016. 1642: p. 197-208.

  1. The sample size of Figure 2B, 3C, 6B, 7B, 7C, 8C, 9B, and 9D should be clarified.

The corresponding indications for sample size have been now included in the corresponding figure note. 

  1. The scale bar in Figure 5B is missing.

Scale bar was added.

  1. Figure6 B and C illustrate the same information. Instead of the percentage of the cell, cell counts in the biomaterials and the spinal cord can demonstrate more information of iNPC survival and proliferation besides its distribution.

Figure 6 is intended to evaluate whether the different versions and components of the implanted biomaterials would influence on the migration/integration of the iNSC into the spinal cord tissue but it was not intended the comparative study of cell survival depending on the different biocomposite version. We have counted cells in both, in the material and in the tissue in order to match both data. We have normalized to 100% the total number of cells found per sample, showing the percentage of those founded cells distributed into both, spinal cord tissue and biomaterial. We believe that having both representations, percentage of cells in the biomaterial and the percentage of cells found in the spinal cord tissue, will increase the comprehensiveness of the described conclusions. 

  1. As illustrated in Figure 6C, almost all of the cells migrate to the spinal cord in the HA_PM_iNPCs group without injury. But in Figure 6A, there are many cells in the scaffold. How do the authors explain the conflicts between fluorescence imaging and quantification? In addition, cell migration into the spinal cord is significantly decreased in the HA_PM_iNPC group with injuries. Why does the migration decrease with injury signals?

In all samples we found cell debris and autofluorescence green signal in the biomaterial, as it is appreciable also in the HA_PM_iNPCs group; this signal was excluded from the cell quantifications, only GFP-positive cells overlapping with DAPi (nuclei) signal were quantified. We appreciate the reviewer comments, and we have included a note for clarification in material and methods (lane 187).

The cells migrate in both cases, in injured and non-injured cords, however, the global survival rates of iNPCs was always lower in the injured groups, with lower number of surviving cells in the spinal cord tissue then, we hypothesize that the cells that migrate into the injured samples died faster than those migrating into the non-injured tissue, and in addition, the cells that still were remanent into the scaffold survived for a longer period. We have included this note in Figure 6 description in lane 446.

  1. In the figure caption of Fig.7A, GFAP is green, not orange.

Thank you, now is corrected

  1. It would be better to show the whole injury area instead of high magnification images in Figure 9A.

Figure 9A was modified as requested.

  1. In Figure 9C, it would be great to have images that demonstrate Neun expression for other treatment groups.

NeuN expression was not significantly different among any of the groups, then, we included an image to graphically describe the parameter that was quantified.

  1. The guidelines of abstracts (Line 41-47) could be removed.

Now is removed

  1. The sentence in Line 119-120 is confusing. It would be great if the authors could clarify which condition, with or without PPY, could help axonal growth?

Now is corrected, in fact, the PPY are the fibers that better help axonal growth as evidenced in reference 24.

Reviewer 3 Report

This is a well-organized and well-illustrated research paper, has an important  message, and should be of great interest to the readers.This paper reported the development of Hyaluronic acid Demilune scaffolds and Polypyrrole Coated Fibers Carrying Embedded Human Neural Precursor Cells and Curcumin for Surface Capping Spinal Cord Injuries.This research opens up new possibilities for treating spinal cord injuries.This manuscript deserves publication after addressing the minor issues cited below.

  1.  Please remove lines 42-47 in the abstract.
  2.  I suggest the authors to describe brielfy about the importance and advantages of "Nano conjugated curcumin" over other forms of curcumin.
  3. Please use the full form of abbreviations during their first usage. For example MTS assay, ki67.   
  4. Did the authors quantify the curcumin concentration in the scafoold? If yes please mention in the methods section
  5. The scale bar text used in the images is not clear. Please increase the size of scale bar text.
  6. I suggest the authors to adjust the contrast of image in Figure 9.c. The image is dull.

Author Response

We really appreciate the very favourable reviewer´s comments.

  1. Please remove lines 42-47 in the abstract.

Removed

  1. I suggest the authors to describe briefly about the importance and advantages of "Nano conjugated curcumin" over other forms of curcumin.

An additional statement including our own reference with the previous description of the synthesis and characterization of the conjugation is now added in the discussion (lane 511)

  1. Please use the full form of abbreviations during their first usage. For example MTS assay, ki67.   

The full names were added

  1. Did the authors quantify the curcumin concentration in the scafoold? If yes please mention in the methods section

No, we did not quantify the amount of Curcumin in the scaffolds, we set the concentration to 5mM in all cases, diluted into the PM from the stock solution.

  1. The scale bar text used in the images is not clear. Please increase the size of scale bar text.

Scale bars edited throughout the manuscript.

  1. I suggest the authors to adjust the contrast of image in Figure 9.c. The image is dull.

Figure 9 C was modified, and contrast was adjusted

Reviewer 4 Report

The novelty of this manuscript is fine, but some important data and figures need to be further provided and refined. Herein, some selective comments towards the authors that might improve this interesting work.

  • Please, check the abstract carefully.
  • Line 207, why did the author use PCL?
  • I really encourage the authors to combine the results with the discussion section.
  • How about the wettability and mechanical properties of the synthesized materials?

Author Response

We appreciate the reviewer´s comments and we have followed and attend the vast majority of reviewer´s requests. In addition, we have attended the requests from other reviewers which include several modifications including several figures. We hope that this reviewer will be in agreement with the improvements included after the global revision.

  • Please, check the abstract carefully.

We have reviewed the entire manuscript with an extensive English editing work, with special attention into the abstract.

  • Line 207, why did the author use PCL?

PCL was used as the abbreviation of poly-ε-caprolactone employed for the Hyaluronic acid demilune scaffold construction as indicated in material and method. We have removed the abbreviation and substitute it for the complete name for clarification. 

  • I really encourage the authors to combine the results with the discussion section.

We appreciate the reviewer´s suggestion however we have decided to separate the description of the results and figures and then focus the discussion in the significance and applicability of the whole obtained results rather than discussion of each experimental data. 

  • How about the wettability and mechanical properties of the synthesized materials?

The fully mechanical characterization of both main components of the implants, the HA scaffold with or without poly-lactic fibers and the PPY fibers were previously published by our group in Doblado LR et al, 2021 and Gisbert Roca et al, 2021 respectively. Both references are included in the material and methods for the description of each material.

Doblado LR, Martínez-Ramos C, García-Verdugo JM, Moreno-Manzano V, Pradas MM. Engineered axon tracts within tubular biohybrid scaffolds. Journal of Neural Engineering 18(4), 0460c0465 (2021)

Gisbert Roca F, García-Bernabé A, Compañ Moreno V, Martínez-Ramos C, Monleón Pradas M. Solid Polymer Electrolytes Based on Polylactic Acid Nanofiber Mats Coated with Polypyrrole. Macromolecular Materials and Engineering 306(2), 2000584 (2021).

Round 2

Reviewer 2 Report

Overall, the authors have addressed the reviewer's comments in a satisfactory manner.